# The SAFFO Study: Sex-Related Prognostic Role and Cut-Off Definition of Monocyte-to-Lymphocyte Ratio (MLR) in Metastatic Colorectal Cancer

**DOI:** 10.3390/cancers15010175

**Published:** 2022-12-28

**Authors:** Camilla Lisanti, Debora Basile, Silvio Ken Garattini, Annamaria Parnofiello, Carla Corvaja, Francesco Cortiula, Elisa Bertoli, Elena Ongaro, Luisa Foltran, Mariaelena Casagrande, Paola Di Nardo, Giovanni Gerardo Cardellino, Gianpiero Fasola, Angela Buonadonna, Nicoletta Pella, Giuseppe Aprile, Fabio Puglisi

**Affiliations:** 1Department of Medical Oncology, Centro di Riferimento Oncologico di Aviano (CRO) IRCCS, 33081 Aviano, Italy; 2Department of Medical Oncology, San Giovanni di Dio Hospital, 88900 Crotone, Italy; 3Department of Oncology, ASUFC University Hospital, 33100 Udine, Italy; 4Sandro Pitigliani Medical Oncology Department, Hospital of Prato, 59100 Prato, Italy; 5Department of Medicine (DAME), University of Udine, 33100 Udine, Italy; 6Department of Medical Oncology, San Bortolo Hospital, Azienda ULSS8 Berica, 36100 Vicenza, Italy

**Keywords:** sex, MLR, colorectal cancer, circulating biomarkers, gender

## Abstract

**Simple Summary:**

In recent years, mounting evidence has recognized the key role of the crosstalk between immune system and cancer cells. Several data have suggested that gender-related immune system composition could impact on both immune response, efficacy of chemotherapy, and immunotherapy and risk of immune-related adverse events. Based on these premises, the present study aimed to evaluate the role of monocyte-to-lymphocyte ratio (MLR), representing the immune suppression cells, in the first place, and immune activating cells, in the second. The analysis, conducted on 490 patients with metastatic colorectal cancer, showed that males and females have a different profile of immune response. Of note, high MLR, both in males and females, is an unfavorable independent prognostic factor.

**Abstract:**

Background: Emerging data suggest that gender-related immune system composition affects both immune response and efficacy of immunotherapy in cancer patients (pts). This study aimed to investigate the sex-related prognostic role of MLR in metastatic colorectal cancer (mCRC) pts. Methods: We analyzed a retrospective consecutive cohort of 490 mCRC patients treated from 2009 to 2018 at the Oncology Departments of Aviano and Pordenone (training set) and Udine (validation set), Italy. The prognostic impact of MLR on overall survival (OS) was evaluated with uni- and multivariable Cox regression models. The best cut-off value to predict survival was defined through ROC analyses. Results: Overall, we identified 288 males (59%) and 202 females (41%); 161 patients (33%) had a right-sided, 202 (42%) a left-sided primary, and 122 (25%) a rectal tumor. Interestingly, gender was associated with MLR (*p* = 0.004) and sidedness (*p* = 0.006). The obtained cut-off value for MLR in females and males was 0.27 and 0.49, respectively. According to univariate analysis of the training set, MLR (HR 9.07, *p* ≤ 0.001), MLR > 0.27 in females (HR 1.95, *p* = 0.003), and MLR > 0.49 in males (HR 2.65, *p* = 0.010) were associated with poorer OS, which was also confirmed in the validation set. In multivariate analysis, MLR > 0.27 in females (HR 2.77, *p* = 0.002), MLR > 0.49 in males (HR 5.39, *p* ≤ 0.001), BRAF mutation (HR 3.38, *p* ≤ 0.001), and peritoneal metastases (HR 2.50, *p* = 0.003) were still independently associated with worse OS. Conclusions: Males and females have a different immune response. Our study showed that high MLR, both in males and females, is an unfavorable Independent prognostic factor. Further prospective studies are needed to confirm these data.

## 1. Introduction

Colorectal cancer (CRC) is the third most common cancer in the world. Despite all available treatment options for metastatic disease, the prognosis of these patients is still poor [1]. One of the most intriguing and novel therapeutic approaches is represented by immunotherapy, particularly with immune checkpoint inhibitors (ICIs), recently approved as a first-line treatment for metastatic colorectal cancer (mCRC) patients with high microsatellite instability (MSI-H) [2].

As is well known, inflammation, immunosurveillance, and immunoediting play a key role in cancer development and spreading [3]. The complex interactions between tumor cells and host immune response are even more interesting considering that a significant proportion of patients have no benefit from anti-cancer treatments. Therefore, it is necessary to identify prognostic and predictive markers that support the clinicians in performing the most accurate patient selection for treatment definition and monitoring of treatment response. Recently, several studies revealed the prognostic value of inflammatory and serum biomarkers, such as neutrophil to lymphocyte ratio (NRL), monocyte to lymphocyte ratio (MLR), lactate dehydrogenase (LDH), and Glasgow prognostic score (GPS), in different cancer types including CRC, at any stage, but they still need to be validated in clinical practice [4,5,6].

Another emerging topic concerns gender-related immune system composition and its impact on both immune response, efficacy of chemotherapy and immunotherapy and risk of immune-related adverse events (irAEs). Many authors highlighted how adult females usually have stronger innate and adaptive immune response compared to males, as demonstrated by an increased prevalence of autoimmune diseases. Conversely, tumors in males are more antigenic with a higher tumor mutational burden (TMB) than in females [7,8,9,10]. Conflicting results about gender-related treatment strategies (immunotherapy +/− chemotherapy) have been reported in different meta-analyses [11,12,13] and recent findings suggest that the role of sex hormones in the immune system, microbiome, and modulation of PD-1/PD-L1 pathway should not be ignored [14,15].

Based on these premises, the purpose of this study is to investigate the gender-related prognostic role of MLR in mCRC patients, defining at the same time a gender-specific cut-off value.

## 2. Materials and Methods

### 2.1. Study Design

The SAFFO study is a retrospective, observational, multicentric study. We retrospectively reviewed and analyzed a cohort of 490 consecutive mCRC patients treated with first-line chemotherapy from 2009 to 2018 at the Oncology Departments of Aviano and Pordenone (training set) and Udine (validation set). Patients with the availability of blood sample analysis were examined according to gender in the SAFFO study, a retrospective, observational, multicentric study. All patients had a histologically confirmed mCRC and had consented for the use of clinical data, rendered anonymous, for purposes of clinical research about epidemiology, training, and the study of diseases. 

Data concerning baseline characteristics were recorded from electronic and paper-based chart review according to strict privacy standards. The study was approved by the Departmental Review Board and by the Ethics Committee CEUR FVG-ARCS (approved in June 2019, Protocol number CRO-2019-28). 

The primary objective of the study was to evaluate the prognostic impact of MLR before the first-line treatment according to gender in mCRC patients, in terms of overall survival (OS). OS was defined as the time between treatment start and death for any cause.

Moreover, the secondary objectives of the study were to identify a threshold able to classify patients according to MLR in a training cohort, and to validate the identified cut-off. 

### 2.2. Blood Sample Analysis

MLR was defined as the absolute monocyte count divided by the absolute lymphocyte count. Full blood count data were eligible for analysis if performed within 1 month before the beginning of first-line chemotherapy (Appendix A). These parameters were analyzed using the results of a peripheral blood cell count performed with a DxH800 Hematology analyzer (Beckman Coulter, Milan, Italy).

### 2.3. Statistical Analysis

The dataset was divided in two cohorts, a training set and a validation set. Patients’ clinic-pathological characteristics were described in the Table 1. Categorical variables were reported through frequency distribution, whereas continuous variables are stated through median range. Association analysis was explored by Chi-squared test or Kruskal–Wallis test, as appropriate. Subgroups of patients identified by MLR according to gender were compared using the log-rank test and considering a *p*-value < 0.05 as statistically significant. The prognostic impact of MLR in the two gender was evaluated in terms of OS with uni- and multivariable Cox regression models, including also potential confounders. The best cut-off value to predict survival was defined through receiver operating characteristic (ROC) analyses. Statistical analysis was performed with STATA (StataCorp. (2015) Stata Statistical Software: Release 14.2. College Station, TX, USA: StataCorp LP) and with MedCalc version 20.210 (MedCalc Software Ltd., Ostend, Belgium).

## 3. Results

Overall, 490 patients treated with a first line chemotherapy for mCRC were included in the present analysis. Of note, 288 (59%) patients were males and 202 (41%) females; 317 (65%) patients were aged ≤70 years. Of note, 161 (33%) patients had a right-sided cancer and 324 (67%) a left/rectum one; a total of 339 (70%) of patients underwent primary tumor resection. Liver was the most frequent metastatic site (36%), followed by peritoneum (22%), lung (19%), lymph-node (14%), bone (3%), and brain (0.6). Approximately 55% of patients had only one metastatic site. Regarding mutational status, *BRAF* and *KRAS* mutations were detected in 40 (8.16%) and 182 (37%) mCRC, respectively. The whole demographic and disease characteristics are listed in Table 1, while patient’s characteristics according to gender are reported in Table 2. The MLR cut-off obtained with ROC analysis was 0.27 (AUC 0.672) in females (Figure 1A,B) and 0.49 (AUC 0.646) in males, according Youden index. Among female patients, 201 (41%) had MLR > 0.27 and 83 (17%) MLR < 0.27, respectively. Among males, 46 (9.5%) had MLR > 0.49 and 155 (32%) MLR < 0.49 respectively. Interestingly, in the whole population, gender was associated with MLR (*p* = 0.004; Figure 2) and sidedness (left/rectum and right *p* = 0.006).

### 3.1. Training Set

Training set population included 263 patients. In a median follow-up of 53.3 months, median overall survival (mOS) was 30.74 months. In univariate analysis, MLR > 0.27 in females (HR 1.95, 95% C.I. 1.25–3.02, *p* = 0.003), MLR > 0.49 in males (HR 2.65, 95% C.I. 1.26–5.59, *p* = 0.010), peritoneal metastases (peritoneal vs. liver: HR 3.05, 95% C.I. 1.98–4.69, *p* < 0.001), BRAF mutation (HR 2.24, 95% C.I. 1.08–4.60, *p* = 0.028), KRAS mutation (HR 1.62, 95% C.I. 1.11–2.34, *p* = 0.011), age ≥ 70 years (HR 1.80, 95% C.I. 1.29–2.52, *p* < 0.001), more than one site of metastasis (HR 2.09, 95% C.I. 1.51–2.91, *p* < 0.001), and sidedness (right vs. left, HR 1.45, 95% C.I. 1.30–1.67, *p* < 0.001) were associated with a poorer OS. Conversely, primary tumor resection (HR 0.40, 95% C.I. 0.25–0.62, *p* < 0.001) was associated with a better OS. In multivariate analysis, MLR > 0.49 in males (HR 4.70, 95% C.I. 1.27–17.42, *p* = 0.020), BRAF mutation (HR 6.53, 95% C.I. 1.98–21.57, *p* = 0.002) and peritoneal metastases (peritoneal vs. liver: HR 5.13, 95% C.I. 1.44–18.28, *p* < 0.012) were associated with a poorer OS. On the contrary, primary tumor resection (HR 0.32, 95% C.I. 0.11–0.90, *p* < 0.032) was associated with a better OS (Appendix A).

### 3.2. Validation Set

Interestingly, to validate results obtained in the training set, 227 patients were included in this validation analysis. At a median follow-up of 55.7 months, mOS was 22.09 months. In univariate analysis, MLR > 0.27 in females (HR 2.21, 95% C.I. 1.21–4.06, *p* = 0.010), MLR > 0.49 in males (HR 2.99, 95% C.I. 1.52–5.90, *p* = 0.002), peritoneal metastases (peritoneal vs. liver: HR 1.88, 95% C.I. 1.25–2.81, *p* = 0.002), lymph-node metastases (lymph-node vs. liver: HR 2.17, 95% C.I. 1.39–3.38, *p* = 0.001), and more than one site of metastasis (HR 1.73, 95% C.I. 1.27–2.36, *p* < 0.001) were associated with a poorer OS. Conversely, primary tumor resection (HR 0.36, 95% C.I. 0.24–0.54, *p* < 0.001) was associated with a better OS (Table 2).

In multivariate analysis, MLR > 0.27 in females (HR 3.31, 95% C.I. 1.29–8.46, *p* = 0.012), MLR > 0.49 in males (HR 8.25, 95% C.I. 2.75–24.67, *p* < 0.001), BRAF mutation (HR 4.72, 95% C.I. 2.01–11.11, *p* < 0.001), peritoneal metastases (peritoneal vs. liver: HR 2.22, 95% C.I. 1.03–4.80, *p* = 0.041), and age ≥ 70 years (HR 1.86, 95% C.I. 1.09–3.15, *p* = 0.021) were associated with a poorer OS. On the contrary, primary tumor resection (HR 0.24, 95% C.I. 0.13–0.45, *p* < 0.001) and more than one site of metastasis (HR 0.45, 95% C.I. 0.22–0.94, *p* = 0.034) were associated with a better OS (Appendix A).

### 3.3. Pooled Population

Finally, the prognostic role of MLR according to gender has been tested in the pooled population. At median follow-up of 55.73 months, median OS was 22.29 months. At univariate analysis, MLR > 0.34 (according to ROC analysis) in overall population (HR 1.83, 95% C.I. 1.46–2.28, *p* < 0.001, Figure 3A), MLR > 0.27 in females (HR 2.07, 95% C.I. 1.48–2.91, *p* < 0.001), MLR > 0.49 in males (HR 2.87, 95% C.I. 1.85–4.45, *p* < 0.001; Figure 3B), peritoneal metastases (peritoneal vs. liver: HR 2.32, 95% C.I. 1.73–3.10, *p* < 0.001), lymph-node metastases (lymph-node vs. liver: HR 2.01, 95% C.I. 1.43–2.83, *p* < 0.001), more than one site of metastases (HR 1.89, 95% C.I. 1.51–2.37, *p* < 0.001), BRAF mutation (HR 1.69, 95% C.I. 1.13–2.51, *p* = 0.009), KRAS mutation (HR 1.37, 95% C.I. 1.08–1.75, *p* = 0.008), age ≥ 70 years (HR 1.51, 95% C.I. 1.20–1.90, *p* < 0.001), and sidedness (HR 1.59, 95% C.I. 1.45–1.76, *p* ≤ 0.001) were associated with a poorer OS. Conversely, primary tumor resection (HR 0.37, 95% C.I. 0.28–0.49, *p* < 0.001) was associated with a better OS. Notably, type of treatment was not associated with survival outcomes.

At multivariate analysis, MLR > 0.27 in females (HR 2.77, 95% C.I. 1.45–5.27, *p* = 0.002), MLR > 0.49 in males (HR 5.39, 95% C.I. 2.50–11.60, *p* < 0.001), BRAF mutation (HR 3.38, 95% C.I. 1.85–6.17, *p* < 0.001), and peritoneal metastases (peritoneal vs. liver: HR 2.50, 95% C.I. 1.36–4.59, *p* = 0.003) were still independently associated with a worse OS. On the contrary, primary tumor resection (HR 0.33, 95% C.I. 0.21–0.53, *p* < 0.001) was associated with a better OS (Table 3). Harrel C statistics was also performed to evaluate if survival analysis about MLR according to gender could have a better performance compared with MLR regardless gender. Harrel C statistics was 0.72 when MLR was evaluated in overall population and 0.73 according to gender evaluation.

## 4. Discussion

Immune suppression and cancer-induced chronic systemic inflammation play a crucial role in paving the way for tumor proliferation and progression [16,17]. Both tumor cells and stromal tumor microenvironment (TME) cells recruit circulating monocytes in the tumor site and promote their differentiation to M2 tumor associated macrophages (TAMs), enhancing immunosuppression [18]. On the other hand, tumor infiltrating lymphocytes predict a better survival [19]. Changes in peripheral blood leucocytes components may mimic the TME immune polarization. Based on these premises, several studies have investigated the prognostic role of inflammatory indexes, such as neutrophil-to lymphocyte ratio (NLR) and lymphocyte-to-monocyte ratio (LMR), but due to their retrospective nature and heterogeneity no definitive conclusion has been drawn [1,5,20,21,22,23]. 

Our recent retrospective study suggested that MLR was an independent prognostic biomarker of worse OS in mCRC (MLR high vs. low: HR 2.15, *p* < 0.001, 95% C.I. 1.47–3.14) [24]. The same findings emerged also in our previous study conducted on elderly mCRC patients treated with first-line chemotherapy, showing that high baseline MLR is an independent prognostic factor in that population (HR 2.99, 95% C.I. 1.68–5.33) in terms of OS [25]. Furthermore, a recent study conducted on 160 CRC resected patients and 42 healthy controls, showed the independent prognostic role of MLR in terms of five-year disease-free survival (HR = 2.903, 95% C.I.: 1.368–6.158, *p* = 0.005) in adjuvant setting [26]. To the best of our knowledge, gender-related differences in peripheral circulating white blood cells and their potential prognostic role in cancer patient have never been evaluated. The present study, conducted on 490 patients, aimed to investigate the gender related prognostic role of a circulating biomarkers, such as MLR, in patients with mCRC. MLR was found to be significantly associated with gender (*p* = 0.04). Considering that for MLR no optimal cut-off has been defined yet, the dataset was divided in two balanced cohorts for each gender to find and validate the optimal prognostic cut-off. In the pooled population, MLR > 0.27 for females (HR 2.21, *p* = 0.010) and MLR > 0.49 for males (HR 2.65, *p* = 0.002) were associated with poorer OS, both at univariate and multivariate analysis. Sex hormones, such as estrogens, progesterone, and androgens, may also influence both adaptive and innate immune systems [26]. Namely, females have a more robust adaptive and innate immune response, leading to a more efficient reaction to infections and, on the other side, being more prone to develop autoimmune disease [27]. Conversely, a higher propensity in males to have a greater recruitment of immunosuppressive cells has been reported. In fact, higher levels of natural killer (NK) cells, macrophages, and neutrophils activity have been described in males [28,29] while antigen presenting cells (APC) are more efficient in females [30]. Even the cytokine production is different: preclinical model showed how TLR-9 ligands stimulate a greater IL-10 production in male, whereas TLR-7 stimulation leads to a greater IFN-α production in females [31,32]. 

Regarding adaptive immunity, females show higher levels of CD4+ T cells and higher CD4+/CD8+ ratios whereas males have higher number of circulatory T regulatory cells and T CD8+ cells [29]. Besides, females’ CD4+ T cells preferentially produce IFNγ (T Helper 1 cytokine) while males’ CD4+ T cells produce higher level of IL-17 (T Helper 2 cytokine) [33]. It is well known that both innate and adaptive immune responses have substantial gender differences. Many genes located on X- and Y-chromosomes modulate the immune system, encoding for transcriptional factors, TLRs, and cytokine receptors, being one factor responsible for the gender oriented immune response [34,35].

Therefore, gender oriented immune response has practical implications, as highlighted by the different response to anti CTLA-4 and anti PD-1 treatment between males and females [11]. The present study, conducted in a real-life cohort of patients with mCRC, is in line with literature data showing that the MLR has a gender related prognostic role, based on the different immune response with higher levels of immunosuppressive cells in males. The above-described sex-dependent pattern of chemokines and cytokines may determine different lymphocytic migration in the TME. Moreover, the Th1 vs. Th2 CD4+ cells orientation may promote monocyte transformation to TAM type 1 or TAM type 2, respectively, explaining how the same circulating MLR confer different prognosis between sexes. 

Despite the innovative results, the present study has some limitations. The first one is represented by its retrospective nature. In addition, our study was not powered to investigate the differences between pre- and post-menopausal women, and hormonal levels were not assessed.

On the other hand, the strengths of the present study are the large cohort of real word patients, the consistency of the known prognostic factors investigated in the multivariate analysis, (i.e., BRAF mutation, MSI status and peritoneal metastases), and the biological rationale underlying our findings.

## 5. Conclusions

The interplay between cancer and the immune system is a complex scenario with many actors on the stage. Furthermore, males and females have a different immune response. The present study revealed the cut-off responsible for predicting poor prognosis, demonstrating that high MLR (MLR > 0.27 for females and MLR > 0.49 for males) is an unfavorable independent prognostic factor. 

Furthermore, to the best of our knowledge, this is the first study to demonstrate a gender dependent MLR prognostic difference, opening new roads towards personalized medicine. However, prospective studies are needed to confirm these data. 

## Figures and Tables

**Figure 1 cancers-15-00175-f001:**
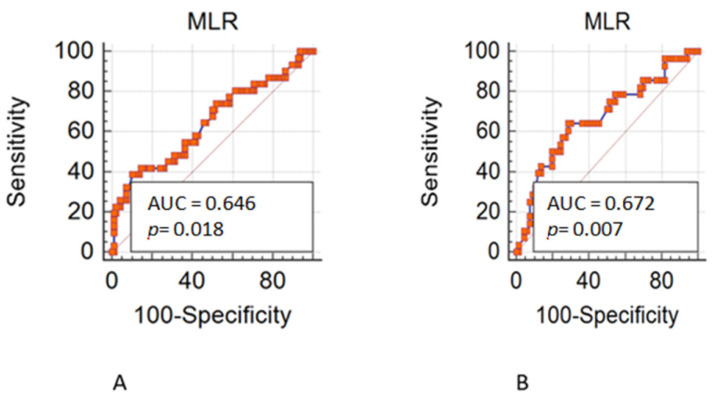
(**A**). Receiver operating characteristic (ROC) in male patients. (**B**). Receiver operating characteristic (ROC) in female patients.

**Figure 2 cancers-15-00175-f002:**
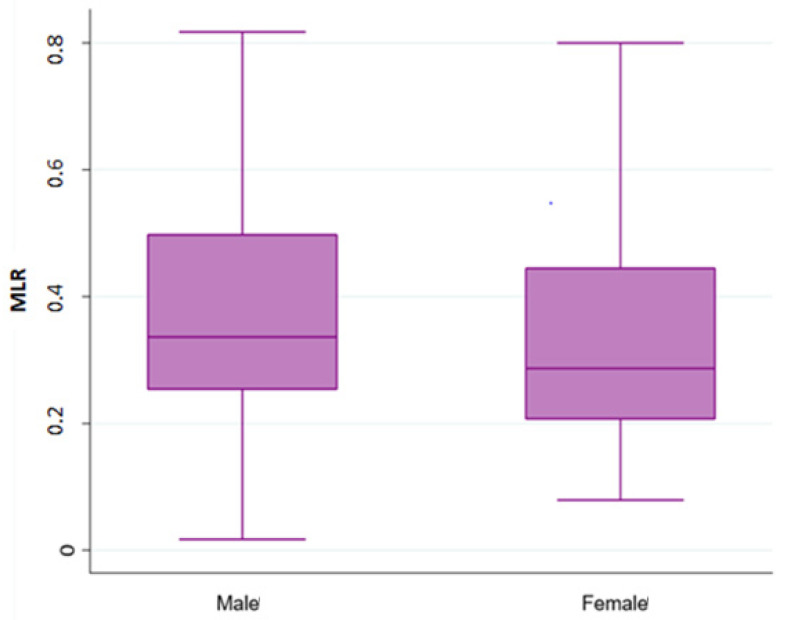
Association analysis between Monocyte-to-lymphocyte ratio (MLR) with gender.

**Figure 3 cancers-15-00175-f003:**
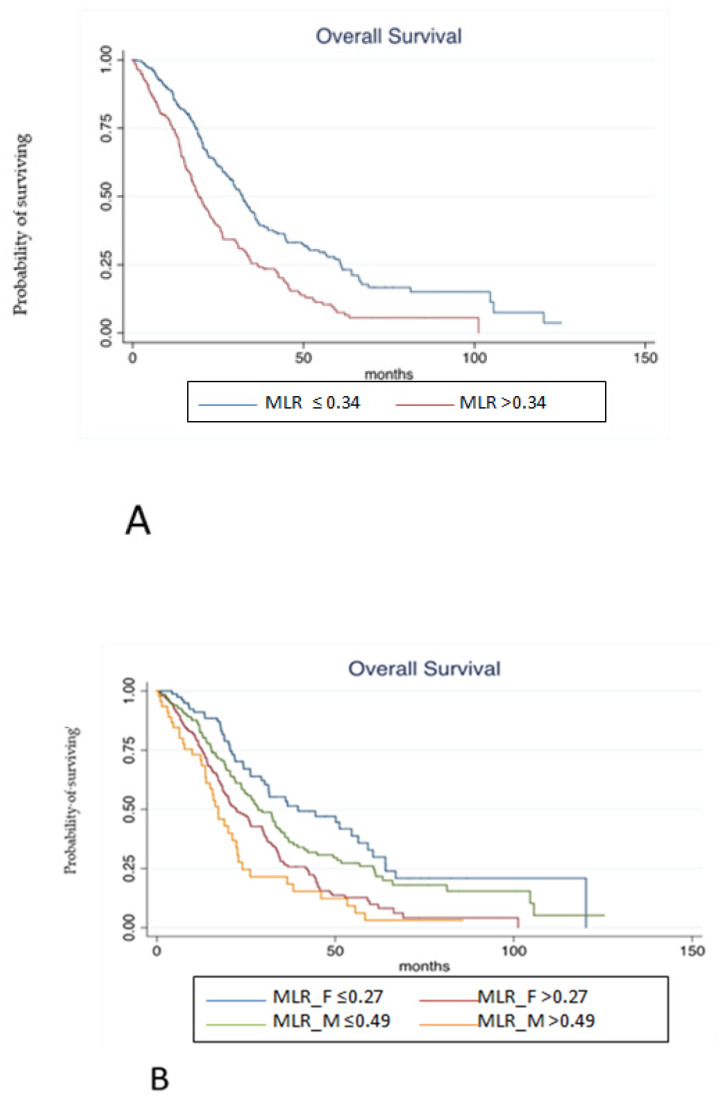
(**A**). Kaplan-Meier estimates of overall survival from random assignment according to MLR in overall population. (**B**). Kaplan-Meier estimates of overall survival from random assignment according to MLR in the distinct gender categories.

**Table 1 cancers-15-00175-t001:** Patient’s characteristics.

Variables	*N* = 490	%
Sex		
M	288	58.78
F	202	41.22
**Age (years)**		
≤70	317	64.69
>70	170	34.69
Missing	3	0.6
**Sidedness**		
Right	161	32.85
Left-Rectum	324	66.12
Missing	5	1.00
**Surgery**		
No	92	18.87
Yes	339	69.18
Missing	59	11.20
**Number of sites**		
1	268	54.69
>1	221	45.10
Missing	1	0.2
**Sites of metastases**		
Liver	178	36.32
Lung	91	18.57
Lymph nodes	70	14.28
Peritoneum	107	21.83
Bone	13	2.65
CNS	3	0.6
Missing	28	5.70
**KRAS**		
WT	239	48.77
Mut	182	37.14
Missing	69	14.08
**BRAF**		
WT	273	55.71
Mut	40	8.16
Missing	177	36.12
Treatment received		
Monotherapy +/− biologic	51	10.41
Doublet	107	21.84
Doublet plus biologic	257	52.45
Triple +/− biologic	75	15.31
**MLR—Training set (N = 263)**		
F ≤ 0.27	61	23.28
F > 0.27	87	33.20
M ≤ 0.49	98	37.40
M > 0.49	16	6.10
Missing	1	0.2
**MLR—Validation set (N = 227)**		
F ≤ 0.27	22	9.69
F > 0.27	114	50.20
M ≤ 0.49	57	25.11
M > 0.49	30	13.21
Missing	4	1.76
**MLR—Overall cohort (N = 490)**		
F ≤ 0.27	83	16.93
F > 0.27	201	41.02
M ≤ 0.49	155	31.63
M > 0.49	46	9.38
Missing	5	1.00

**Table 2 cancers-15-00175-t002:** Patient’s characteristics according to gender.

Variables	Male *N* = 288	Female *N* = 202	*p*-Value
**Age (years)** ≤70 >70 Missing	188 100 0	129 70 3	0.918
**Sidedness** Right Left-Rectum Missing	79 208 1	82 116 1	0.001
**Surgery** No Yes Missing	61 190 37	31 149 22	0.077
**Number of sites** 1 >1 Missing	154 134 0	114 87 1	0.478
**Sites of metastases** Liver Lung Lymph nodes Peritoneum Bone CNS Missing	111 56 42 55 9 3 12	67 35 28 52 4 0 16	0.284
**KRAS** WT Mut Missing	144 108 36	95 74 33	0.850
**BRAF** WT Mut Missing	167 19 102	106 21 75	0.100
**Treatment received** Monotherapy +/− biologic Doublet Doublet plus biologic Triplet +/− biologic	29 66 144 49	22 41 113 26	0.461

**Table 3 cancers-15-00175-t003:** Uni- and multivariate overall survival analyses in pooled population.

	Univariate Analysis			Multivariate Analysis		
Variables	HR	*p*-Value	95% C.I.	HR	*p*-Value	95% C.I.
**MLR—Overall cohort** F > 0.27 M ≤ 0.49 M > 0.49	2.07 1.38 2.87	≤0.001 0.073 ≤0.001	1.48–2.91 0.97–1.97 1.85–4.45	2.77 1.85 5.39	0.002 0.068 ≤0.001	1.45–5.27 0.95–3.58 2.50–11.60
**KRAS** Mut vs. wt	1.37	0.008	1.08–1.75	1.19	0.392	0.79–1.82
**BRAF** Mut vs. wt	1.69	0.009	1.13–2.51	3.38	≤0.001	1.85–6.17
**Sidedness** Right vs. Left	1.59	≤0.001	1.45–1.76	1.20	0.373	0.80–1.79
**Number of sites** >1 vs. ≤1	1.89	≤0.001	1.51–2.37	0.70	0.184	0.41–1.18
**Treatment received** Monotherapy +/− biologic Doublet Doublet plus biologic Triplet +/− biologic	1 1.16 0.77 0.73	0.471 0.174 0.169	0.77–1.74 0.53–1.12 0.47–1.14			
**Sites of metastases** Lung vs. liver Lymph nodes vs. liver Peritoneum vs. liver Bone vs. liver CNS vs. liver	1.05 2.01 2.32 1.57 42.73	0.720 ≤0.001 ≤0.001 0.243 ≤0.001	0.77–1.45 1.43–2.83 1.73–3.10 0.73–3.39 12.74–143.32	0.98 1.12 2.50 1.10 30.98	0.966 0.738 0.003 0.832 ≤0.001	0.54–1.79 0.55–2.27 1.36–4.59 0.43–2.83 6.22–154.13
**Surgery** Yes vs. No	0.37	≤0.001	0.28-0.49	0.33	≤0.001	0.21–0.53

Notably, high MLR (defined as MLR > 0.27 in females and MLR > 0.49 in males) was more frequently found in females than in males (41% vs. 9%).

## Data Availability

All data generated during this study are included in this article. Further enquiries can be directed to the corresponding author.

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
