# Peer review of "The SAFFO Study: Sex-Related Prognostic Role and Cut-Off Definition of Monocyte-to-Lymphocyte Ratio (MLR) in Metastatic Colorectal Cancer"

_cancers, 2022, doi:10.3390/cancers15010175_

Round 1

Reviewer 1 Report

The manuscript by Lisanti et al entitled” The SAFO study: Sex-related prognostic role and cut-off definition of monocyte-to-lymphocyte ratio (MLR) in metastatic colorectal cancer” describes a retrospective analysis of monocyte and lymphocyte total count data from blood samples from a large metastatic colorectal cancer cohort. The analyses distinguish male and female and identify cut-off MLR ratios for each which then are applied to demonstrate prognostic value of MLR for overall survival.

There have been a number of similar studies published by the authors and independent groups, all finding the MLR being a prognostic marker. However, this studies patient cohort is larger and importantly this study is the first to analyse the data for male and female separately and thereby identifying the different cut-offs optimal for overall survival prognosis. Sex differences in the immune system and response are gaining strong interest in recent years and little is known so far about this in the context of the tumor immune environment, making the observations from this manuscript impactful. However, the manuscript appears incomplete and prepared at the quality required for publication. In addition, there are a number of major and minor concerns and questions this reviewer wants to raise with the authors:

Major:  

1) Incomplete and poor structure:

Figure legends are mission entirely,

Figure 2 is not referenced in the text and has not been described in the text

Figures must be referenced in the text in order (first cited figure is Fig 3 at the moment)

2) Evidence/analyses comparing overall prognostic value of MLR  versus female and male separated: data of  MLR’s (of female and male together) prognostic value should be shown and then analysis performed to demonstrate that separating female and male is better!

3) Confounder analysis/consideration: While mentioned in the introduction, this should be more addressed with analysis and in the discussion. Correlation analysis of gender vs all Clinicopathological parameters for example would be good. “only” OS data is shown, it would be good to also perform RFS analysis. Also, the treatment the patients undergo after blood collection, will have to be considered further.

4) Text/language/ style:  Overall the text must be improved. For example, each subsection of the results should have an introductory sentence. The discussion mentions relevant literature but does so in a summarising fashion. The referenced findings must be compared and related directly to the authors own findings.  Improve clarity of wording (i.e. difficult to understand what is meant, in discussion bottom paragraph: “Gender oriented immune response is not just an end in itself, but may have practical implications, as….”).

5) Table and Figure presentation is poor/ incompletely labelled: 

Table1: Something is incorrect here: as % values for KRAS and BRAF are the same, but the patient numbers are not!

Table2: it must be made clear that the right-hand site is the multivariant analysis by labelling. Why are some multivariant analysis values missing?

Table2: Also provide complete data from training set and validation set, currently some values are only mentioned in the text!

Table2: Sidedness data do not match values in the text!

Fig2: Y axis label missing

It would be very nice to show more Kaplan Meier graphs (MLR-OS together and female male separated (of is that Fig2? Can’t tell due to lacking figure legends and

6) References:

Ref 1 is not appropriate for the statement! But should be mentioned later with Ref 4-6

Ref 2 is a very nice review but from 2010, a lot of newer data and reviews are available

A few very relevant studies have not been cited i.e. Jakubowska et al 2020, WJG (DOI: 10.3748/wjg.v26.i31.4639)

Minor:

1 )Would be nice to include a graph presenting the total monocyte and lymphocyte counts and also define parameters of exclusion of values (i.e to low or to high values, which maybe associated with other pathologies).

2) More detail for the ROC analysis could be provided, at least the full name of the analysis should be provided once in the text and in the figure legend.

3) These authors define MLR but other calculate LMR, which makes it harder to compare studies to each other. (This reviewer does recognize that this is an issue of the field)

Author Response

Dear Editor,

We would like to thank you and the reviewers for the revision of our manuscript.

We appreciated the reviewers’ constructive comments and we revised the manuscript accordingly.

Please find our point-by-point response to the reviewer comments. Changes are tracked in the text (red color).

Reviewer 1:

Comments to the Author:

The manuscript by Lisanti et al entitled” The SAFFO study: Sex-related prognostic role and cut-off definition of monocyte-to-lymphocyte ratio (MLR) in metastatic colorectal cancer” describes a retrospective analysis of monocyte and lymphocyte total count data from blood samples from a large metastatic colorectal cancer cohort. The analyses distinguish male and female and identify cut-off MLR ratios for each which then are applied to demonstrate prognostic value of MLR for overall survival.

There have been a number of similar studies published by the authors and independent groups, all finding the MLR being a prognostic marker. However, this studies patient cohort is larger and importantly this study is the first to analyse the data for male and female separately and thereby identifying the different cut-offs optimal for overall survival prognosis. Sex differences in the immune system and response are gaining strong interest in recent years and little is known so far about this in the context of the tumor immune environment, making the observations from this manuscript impactful. However, the manuscript appears incomplete and prepared at the quality required for publication. In addition, there are a number of major and minor concerns and questions this reviewer wants to raise with the authors:

Major:  

1) Incomplete and poor structure:

Figure legends are mission entirely,

Figure 2 is not referenced in the text and has not been described in the text

Figures must be referenced in the text in order (first cited figure is Fig 3 at the moment)

We thank the reviewer for these remarks. Figures have been cited, highlighted and described as requested.

2) Evidence/analyses comparing overall prognostic value of MLR  versus female and male separated: data of  MLR’s (of female and male together) prognostic value should be shown and then analysis performed to demonstrate that separating female and male is better.

According to reviewer’s suggestion, data on MLR’s prognostic value in the overall population have been implemented in 3.3 section and showed in Figure 3A. Moreover, Harrel C statistics was performed to evaluate if survival analysis about MLR according to gender could have a better performance compared with MLR regardless gender. Harrel C statistics was 0.72 when MLR was evaluated in overall population and 0.73 according to gender evaluation.

3) Confounder analysis/consideration: While mentioned in the introduction, this should be more addressed with analysis and in the discussion. Correlation analysis of gender vs all Clinicopathological parameters for example would be good. “only” OS data is shown, it would be good to also perform RFS analysis. Also, the treatment the patients undergo after blood collection, will have to be considered further.

We thank the reviewer for the comment. Correlation analysis of gender vs clinic-pathological features has been performed and reported in table 2. We agree that RFS and/or PFS analysis could add interesting information; unfortunately, due to retrospective design of the study, PFS data are missing or unreable. It is our goal to evaluate the impact of MLR according to gender on PFS in a future evolution of this study. Information about treatments have been described in the table 1 and table 2. Moreover, cox-regression analysis was also performed and listed in table 3.

4) Text/language/ style:  Overall the text must be improved. For example, each subsection of the results should have an introductory sentence. The discussion mentions relevant literature but does so in a summarising fashion. The referenced findings must be compared and related directly to the authors own findings.  Improve clarity of wording (i.e. difficult to understand what is meant, in discussion bottom paragraph: “Gender oriented immune response is not just an end in itself, but may have practical implications, as….”).

We thank the reviewer for the observation. The text has been updated accordingly.

5) Table and Figure presentation is poor/ incompletely labelled: 

Table1: Something is incorrect here: as % values for KRAS and BRAF are the same, but the patient numbers are not!

Table2: it must be made clear that the right-hand site is the multivariant analysis by labelling. Why are some multivariant analysis values missing?

Table2: Also provide complete data from training set and validation set, currently some values are only mentioned in the text!

Table2: Sidedness data do not match values in the text!

Fig2: Y axis label missing

It would be very nice to show more Kaplan Meier graphs (MLR-OS together and female male separated (of is that Fig2? Can’t tell due to lacking figure legends and

Thank you for your comments. Table and figure labels have been improved as suggested.

Moreover:

  • Table 1: it has been updated with right percentage values
  • Table 2 (table 3 of the revised version): univariate and multivariate analyses have been correctly labeled. The missing values of multivariate analysis have been reported.
  • Table 2 (table 3 of the revised version): values about training and validation set were reported in supplementary tables 1 and 2.
  • Table 2 (table 3 of the revised version): sidedness data have been updated in text according to values reported in the table.
  • Fig 2 (figure 3 of the revised version): label of y axis has been reported as suggested
  • Kaplan-Meier graph about MLR OS of overall population has been implemented (Figure 3A)

6) References:

Ref 1 is not appropriate for the statement! But should be mentioned later with Ref 4-6

Ref 2 is a very nice review but from 2010, a lot of newer data and reviews are available

A few very relevant studies have not been cited i.e. Jakubowska et al 2020, WJG (DOI: 10.3748/wjg.v26.i31.4639)

We thank the reviewer for this suggestion.

  • Ref 1 was changed with a more appropriate one.
  • Ref 2 (ref 3) has been changed with a more recent review about inflammation, immunity and cancer.
  • Studies, such as Jakubowska et al 2020, WJG have been added and discussed.

Minor:

  • Would be nice to include a graph presenting the total monocyte and lymphocyte counts and also define parameters of exclusion of values (i.e to low or to high values, which maybe associated with other pathologies).

A graph reporting the total lymphocyte and monocyte count has been added as supplementary figure 1 and described in “Methods” section. No parameters of exclusion of values has been used.

  • More detail for the ROC analysis could be provided, at least the full name of the analysis should be provided once in the text and in the figure legend.

Thank you for your remark. Accordingly, more details about ROC analysis have been added and the full name was provided.

  • These authors define MLR but other calculate LMR, which makes it harder to compare studies to each other. (This reviewer does recognize that this is an issue of the field)

We agree with this observation. However, our study has the main objective to demonstrate the prominent role of the immunosuppressive features. Therefore, we have chosen to shed light on the other side of the coin, rather than underlining the role of lymphocytes. In fact, MLR better describes the immunosuppressive role of monocytes, cwhereas LMR focuses on the pro-immune activity of lymphocytes.

Reviewer 2 Report

The SAFO study offers an interesting insight in the role of MLR in colorectal cancer and the role of gender in the tumor immunological response.

The article is well written, however, I recommend spell checking the paper. 

Author Response

Dear Editor,

We would like to thank you and the reviewers for the revision of our manuscript.

We appreciated the reviewers’ constructive comments and we revised the manuscript accordingly.

Please find our point-by-point response to the reviewer comments. Changes are tracked in the text (red color).

Reviewer 2:

The SAFFO study offers an interesting insight in the role of MLR in colorectal cancer and the role of gender in the tumor immunological response.

  • The article is well written, however, I recommend spell checking the paper. 

We thank the reviewer for these observations. The spell checking has been performed as suggested.